# Ovarian and Energy Status in Lame Dairy Cows at Puerperium and Their Responsiveness in Protocols for the Synchronization of Ovulation

**DOI:** 10.3390/ani13091537

**Published:** 2023-05-04

**Authors:** Anastasia Praxitelous, Panagiotis D. Katsoulos, Angeliki Tsaousioti, Christos Brozos, Ekaterini K. Theodosiadou, Constantin M. Boscos, Georgios Tsousis

**Affiliations:** 1Clinic of Farm Animals, School of Veterinary Medicine, Faculty of Health Sciences, Aristotle University of Thessaloniki, 54627 Thessaloniki, Greece; katsoulo@vet.auth.gr (P.D.K.); tsaoange@vet.auth.gr (A.T.); brozos@vet.auth.gr (C.B.); pboscos@vet.auth.gr (C.M.B.); tsousis@vet.auth.gr (G.T.); 2Department of Physiology, Faculty of Veterinary Science, School of Health Sciences, University of Thessaly, 43100 Karditsa, Greece; etheodosiadou@uth.gr

**Keywords:** lameness, dairy cattle, ovarian resumption, energy status, synchronization, fertility

## Abstract

**Simple Summary:**

Lameness is a prevailing problem in dairy herds that has a negative effect on welfare, reproduction, and economic viability. The aim of this study was to investigate if the negative impact of lameness on ovarian activity at the end of puerperium is related to the energy status of dairy cows and to measure the responsiveness of lame dairy cows to hormonal manipulations. Lame cows had poorer ovarian activity at the end of puerperium; however, this finding was not associated with the energy status of the cows. Most lame cows responded well to estrous synchronization protocols and showed adequate long term reproductive performance. The severity of lameness was explanatory mainly for the risk of a cow to be culled during the study, whereas the type of lesion was primarily associated with lower fertility. Our results emphasize the need for the prompt diagnosis and treatment of lame cows. Additionally, these cows require intensive reproductive management due to the greater risk of being anovulatory at puerperium.

**Abstract:**

The purpose of this study was to assess the ovarian and energy status of multiparous lame dairy cows at the end of puerperium and investigate their responsiveness to estrous synchronization treatment regimens. Initial lameness scoring was performed at 28 ± 5 and 37 ± 5 d post partum, followed by lesion documentation and treatment. Cows were blocked by lameness severity and were randomly allocated to an estrous synchronization treatment regimen with seven days of progesterone supplementation (group LP, *n* = 26) or with an administration of PGF_2α_ twice, 14 d apart (group LC, *n* = 26). Non-lame cows served as controls (group C, *n* = 27) and the same treatment regimen was imposed as that for group LC. Twelve days after estrous presynchronization, an Ovsynch treatment regimen and timed AI were imposed. Ultrasonography of the ovaries and blood sampling for progesterone were used to assess cyclicity status, whereas β-hydroxybutyrate (BHBA) and non-esterified fatty acids (NEFA) were used to assess energy status. Lame cows were to a greater proportion non-cycling (36.5% vs. 11.1%; *p* = 0.02), had greater overall NEFA concentrations (0.32 ± 0.02 vs. 0.26 ± 0.02 mEq/L; *p* = 0.02) and a greater incidence of elevated NEFA concentrations (53.9% vs. 29.6%, *p* = 0.04) compared to control cows. However, no interaction between energy and lameness status was evident regarding non-cycling cows. The percentage of cows responding to the presynchronization, synchronization and ovulating did not differ between groups LP, LC, and C. The first-service conception rate (FSCR) tended to be greater for group C (37.0%) compared to group LP (16.0%; *p* = 0.08). Long-term reproductive performance did not differ between lame and control cows, although culling rates did (21.2% vs. 0%, respectivly; *p* = 0.01). The severity of lameness had an effect on culling rates (30.6% vs. 0% for cows with marked vs. moderate lameness; *p* = 0.01), whereas the type of lesion largely explained poor reproductive performance (FSCR 13.9% vs. 40.0% for cows with claw horn disruptions vs. infectious lesions; *p* = 0.04). Conclusively, cows that were lame during puerperium are at a greater risk of not cycling irrespective of energy status. Treatment regimens for the synchronization of ovulation seem to be efficient at resuming ovarian cyclicity. Marked lameness was detrimental to survivability, whereas cows with claw horn lesions had compromised reproductive capacity.

## 1. Introduction

Lameness is a major concern of the dairy industry, since it has detrimental effects on the reproductive performance of cows that are brought on through multiple pathophysiological mechanisms and affects economic viability. In various studies [1,2], the pregnancy rates of lame cows were markedly lower compared to those of non-lame herd mates.

Lameness is an acute or chronic stressor, which can have detrimental effects on endocrine function and impair reproductive performance through actions at the hypothalamic–pituitary–adrenal axis [3]. The adrenocorticotrophic hormone suppresses the pulsatile secretion of LH and induces persistent follicles and delayed ovulation [4]. Additionally, the administration of exogenous glucocorticoids suppresses pituitary gonadotropin secretion, disrupts metabolic signaling [5] and decreases progesterone secretion from the corpus luteum (CL) [6]. Consequently, lame cows compared to non-lame ones have a delayed resumption of ovarian cyclicity post partum [7,8], a greater incidence of cystic ovarian disease [9], lower progesterone concentrations [8,10,11], and greater embryo losses [12]. 

Furthermore, lameness is an indirect suppressor of fertility in dairy cows, due to physical limitations. Estrous expression is less intense [13,14], and lame cows are mounted less frequently [13,15] and for shorter periods [10] compared to non-lame cows. 

One way to potentially bypass both hormonal and physical restrictions is to impose estrous synchronization regimens on lame cows. However, only in a few studies has this hypothesis been tested. Ingenhoff et al. [16] investigated the effect of lameness at the onset of an Ovsynch protocol in non-pregnant cows and found that it decreased the odds of pregnancy. McNally et al. [1] imposed an estrous synchronization regimen on lame cows using a GnRH-PRID-PGF_2α_ protocol, and although estrous expression did not differ between lame and non-lame cows, conception rates were compromised in lame cows. Nevertheless, to our knowledge, the exact ovarian response of lame cows to estrous synchronization treatment regimens has not been previously reported. 

Additionally, cows with severe lameness devote less time feeding [17,18,19] and have less dry matter intake compared to non-lame cows [20]. It is hypothesized that lameness, due to this altered feeding behavior, can result in a greater negative energy balance (NEB) and greater body condition score (BCS) lowering. In the recent research of Daros et al. [21], lameness was associated with reduced feeding time, which in turn was associated with an increased likelihood of subclinical ketosis, though the BCS was not affected. Nevertheless, the disease route can also be reversed. Alawneh et al. [22] reported that the body weight of dairy cows decreased during a period of three weeks before lameness was diagnosed and for a period of as long as 4 weeks post-treatment. As the BCS decreases, due mainly to NEB, the prophylactic digital cushion of the hoof weakens, which predisposes cows to sole ulcers and white line disease [23,24]. Cows with less-than-optimal body conditions are at greater risk of developing lameness [25,26]. Furthermore, NEB has been confirmed to suppress reproductive performance, because it increases the risk of anovulation post partum [27,28,29], delays the onset of ovarian functions [30] and negatively affects a wide range of reproductive variables [31,32,33]. However, the direct linkage between lameness, metabolic and ovarian status in dairy cows is relatively unexplored [34]. 

The primary aim of the present study was to assess the ovarian and energy status of multiparous lame cows at the end of the puerperal period and the onset of the breeding period and to investigate associations with reproductive performance. We also hypothesized that using a progesterone-based presynchronization treatment regimen to induce the onset of estrous cycles would mitigate to some extent the negative consequences of lameness on specific reproductive variables. Furthermore, the effect of different types and the severity of lameness on reproductive variables was analyzed retrospectively. 

## 2. Materials and Methods

### 2.1. Ethical Statement

This study was approved by the Assembly of the Faculty of Veterinary Medicine, Aristotle University of Thessaloniki (69/30 June 2016). All procedures complied with the EU Directive 2010/63/CE. 

### 2.2. Animals and Housing

The study was conducted on three dairy farms with Holstein Friesian cows located in the region of Central Macedonia, Greece, around Lake Koroneia from July 2016 to December 2019. At the initiation of the experiment, the average herd size was 300 cows with an average milk production per cow per year of 10,259 kg. Animals were housed indoors in free-stall barns with concrete floors, automatic scraper systems and cubicles with mattresses. Animals were fed twice daily total mixed rations formulated to meet energy requirements according to the NRC recommendations (2001) [35] and had free access to water. 

### 2.3. Exclusion Criteria

All cows (*n* = 79) enrolled in this study were multiparous, to exclude first-parity effects on reproduction [36]. Initially, all cows were examined within 24 h subsequent to parturition. Animals with dystocia, peri-parturient (retained placenta, and milk fever) or other (mastitis, abomasum displacement, and severe lameness) diseases were excluded from the study. Eligible cows were examined clinically on a weekly basis until the onset of the experimental procedure, i.e., at 28 ± 5 days post partum (d p.p.), and the same exclusion criteria were applied.

### 2.4. Experimental Design

All interventions (I1-8) are presented in Figure 1.

I1: At 28 ± 5 d p.p., there was lameness scoring (LSC) using a standardized 5-point scale system proposed by Sprecher et al. [37] for all eligible cows. Lame cows were blocked by severity (scoring 3 or 4) and were randomly allocated to two estrous presynchronization treatment regimens (group LP: *n* = 26; LC: *n* = 26) that were initiated at I2. Cows with severe lameness (LSC 5) were excluded from the study for ethical reasons and were treated promptly. Additionally, there was an assignment to a control group of non-lame cows (group C, LSC 1, and *n* = 27). Cows with LSC 2 in I1 were excluded from the study. Every triplet of cows (LP, LC and C) was enrolled from the same farm and within a maximum of one month, to account for uniform management procedures and conditions. In all cows, transrectal ultrasonography of the ovaries (U/S) and blood sampling (BS) were performed. 

I2: At 37 ± 5 d p.p., U/S, BS and LSC were repeated. Examination and trimming of all claws were performed in a chute and lesions were recorded. The appropriate therapy according to standard principles [38] was initiated, including the use of blocks, bandaging, nonsteroidal anti-inflammatory drugs, and topical antibiotics in a spray form. The following estrous presynchronization treatment regimens were subsequently initiated.

Group Lame-Progesterone (LP; *n* = 26): Lame cows were administered a progesterone-releasing intravaginal device containing 1.55 g of progesterone (PRID^®^ Delta, Ceva Santé Animale, Loudéac, France) after disinfection of the vulva with a mild (5%) povidone–iodine (Betadine^®^, Lavipharm, Athens, Greece) solution. 

Group Lame-Control (LC; *n* = 26): Lame cows were administered an intramuscular injection of 500 µg of cloprostenol (a synthetic Prostaglandin F_2α_ analogue and 2 mL of Estrumate^®^, MSD Animal Health, Boxmeer, The Netherlands). 

Group Control (C, *n* = 27): The non-lame cows were administered PGF_2α_, following the imposition of the estrous presynchronization treatment regimen of group LC. 

I3: At 44 ± 5 d p.p., the PRID device was removed and a single administration of PGF_2α_ was given to the cows of the LP group, whereas at 51 ± 5 d p.p., groups LC and C received a second administration of PGF_2α_, completing the imposition of the estrous presynchronization regimens. Subsequently, all groups received the same estrous synchronization treatment regimen (Ovsynch) for timed AI (TAI).

I4: Twelve days after the end of the presynchronization treatment regimen, 0.01 mg of buserelin (GnRH agonist; 2.5 mL of Receptal^®^, MSD Animal Health, Boxmeer, The Netherlands) was administered intramuscularly to all cows, followed by conducting U/S, BS and LSC procedures. 

I5: Seven days later, cows were treated with PGF_2α_, and U/S and BS procedures were performed.

I6: After 56 h, there was a second GnRH administration, and U/S, BS and LSC procedures were performed.

I7: Twelve to sixteen hours later, there was TAI and U/S procedures were performed. 

I8: Blood sampling was performed 12 days post insemination (p.i.). 

Pregnancy was diagnosed by U/S at 40 days p.i. Cows detected in estrus before pregnancy diagnosis were artificially inseminated through the cervix. After a negative pregnancy diagnosis, the cows were enrolled in the reproductive management protocols applied to each farm and all information regarding inseminations and outcomes were retrieved from farm software. All samplings, treatments and examinations were conducted by the first author of this manuscript, except for TAI. 

### 2.5. Clinical Assessment of Lameness 

Lesions identified were attributed to either claw horn disruptions (CHDL) or infectious diseases (ID). Claw horn disruptions (*n* = 36) included sole and bulb ulcers, white line disease lesions, diffused sole/bulb hemorrhages and laminitis. Interdigital dermatitis and digital dermatitis were classified as ID (*n* = 15) [39,40]. One case was mixed and was classified as CHDL. The number of affected claws was recorded. Cows with other lameness aetiology (i.e., arthritis, acute injuries or unspecified) were excluded from the study. Cows assigned to group C had no lesions. 

The severity of lameness was based on LSC. Specifically, every case with two consecutive (at I1 and I2) lameness scores of 4 or one lameness score of 4 and concurrent lesions in more than one foot was characterized as markedly lame (*n* = 36). All other cases were of moderate severity (*n* = 16). Severity scores did not differ between groups LP and LC (with 18 marked and 8 moderate cases in each group). 

### 2.6. Ultrasonography

At the onset of the experimental procedures (28 ± 5 d p.p.), vaginoscopy, transrectal palpation and B-mode sonography (Honda HS 101V with a 5 MHz linear transducer) were used to determine the presence of abnormal content in the uterus and vagina and evaluate the involution of the uterus (horn symmetry and repositioning to the pelvic canal). Cows with pathological findings of the uterus (incomplete uterine involution, hyperechogenic intrauterine or intravaginal content) were excluded from the study. The ovarian structures were located, and images of the ovaries where structures (CL, follicles, and ovarian cysts) were the greatest in diameter were frozen and stored for analysis with appropriate software (Inkscape^®^, New York, NY, USA).

### 2.7. Reproductive Definitions and Outcomes

Ovarian mapping and concurrent progesterone concentrations (P4, ng/mL) were evaluated to define reproductive outcomes. Resumption of ovarian functions was confirmed at I1 and I2 for cows with a detectable CL and P4 > 1 ng/mL in at least one of the two examinations (*n* = 57/79). Any other condition was classified as non-cyclic (*n* = 22/79) and concerned three types of atresia with low P4 (≤1 ng/mL): the development of cystic ovarian disease, i.e., presence of an ovarian structure of > 25mm with an absence of a CL (*n* = 8), repeated follicular waves with no ovulations from a dominant follicle (*n* = 12) and two cases of anovulation with failure to develop a follicle of ≥10 mm [41,42,43].

Presynchronization success in inducing ovarian functions (*n* = 59/79) was defined as a cow with the presence of an active CL (P4 > 1 ng/mL) at the onset of imposing the estrous synchronization treatment regimen (I4). Estrous synchronization success (*n* = 71/79) was defined as a cow with the presence of a dominant ovarian follicle (diameter ≥ 10 mm) in I6, combined with either a regression of the CL (decrease in P4 between I5 and I6 of > 75%) or no active CL at I5 and I6 (P4 ≤ 1 ng/mL). A cow with a dominant follicle at the time of conducting TAI and greater than the minimal threshold concentration of P4 12 d later (I8), followed by normal estrous intervals or a positive pregnancy diagnosis, was considered to have ovulated as a result of imposing the treatment regimens. 

### 2.8. Blood Sampling and Analytic Assays

Blood sampling was always performed after morning milking and before feeding. Blood was collected by coccygeal venipuncture into 10 mL vacuum polyethylene tubes without an anticoagulant (BD Vacutainer^®^, Becton, Dickinson and Company, Franklin Lakes, NJ, USA). The samples were stored in a refrigerator and transferred in a cooler to be centrifuged (1006× *g* for 20 min) approx. 3 h after collection. Serum was transferred into Eppendorf-type tubes and stored at −20 °C until analysis. Blood samples from interventions 1–6 and 8 were used for the detection of P4. Serum P4 concentration was determined, in duplicate, using solid-phase RIA procedures (gamma counter Wizard 1480, PerkinElmer, Turku, Finland) with a commercially available radioimmunoassay kit (IMMUNOTEC^®^, Prague, Czech Republic). The smallest detection limit was 0.03 ng/mL. The intra- and inter-assay coefficients of variation were <10%. A cow with serum P4 concentration of ≤1 ng/mL was documented as having no active luteal tissue. 

Blood samples from interventions 1, 2 and 6 were used to investigate the metabolic profile of the experimental cows, through the detection of ß-hydroxybutyric acid (ΒHΒA) and non-esterified fatty acids (NEFA). The cut off for greater-than-optimal BHBA (BHBA+) was set at 1.2 mmol/L [44] and that for greater-than-optimal NEFA (NEFA+) was set at 400 mEq/L [45]. The analytic method utilized was spectrophotometry (Pentra C400^®^, HORIBA ABX SAS, Montpellier, France). The intra- and inter-assay coefficients of variation for all these analyses were <10%. 

### 2.9. Statistical Analysis 

Data were analyzed utilizing the online platform SAS^®^ OnDemand for Academics (SAS Institute, Cary, NC, USA). The sample size was initially estimated for continuous variables (BHBA, NEFA, and P4). A difference regarding NEFA concentrations between control and lame cows, similar to those found on day 28 (0.29 vs. 0.37 with a standard deviation of 0.12), demanded a total sample size of 60 cows for the test power to exceed 0.80 and a total of 100 cows for the test power to reach 0.95. However, the test power hardly exceeded 0.80 for binary variables. Continuous variables (BHBA and NEFA values, number of AIs per pregnancy, and days open) were tested for normality with the Shapiro–Wilk test. When there was a normal distribution of data (BHBA, NEFA), simple comparisons between independents were performed using Student’s *t*-test, whereas repeated measurements in time were analyzed using general linear mixed models. Wilcoxon’s signed-rank test and the Kruskal–Wallis one-way analysis of variance were used to compare independent groups when there were non-normal data distributions. Proportions were compared using the Chi-squared test and Fisher’s exact tests. The generalized linear mixed model was used to investigate possible interactions between ovarian and metabolic status using reproductive indices. The rate of becoming pregnant for control and lame cows was evaluated using the Kaplan–Meier survival curves. Depending on the statistical procedure used, data are presented as least-square means, arithmetic means, medians, or proportions, and are clearly stated in the text. It was considered that there were differences when there was approximately a p value of < 0.05. Exact *p*-values are provided in the Appendix A. 

## 3. Results

### 3.1. Metabolic Profile of Lame Cows

There was no difference between lame and control cows regarding BHBA concentrations overall (0.67 ± 0.05 vs. 0.68 ± 0.07, *p* = 0.91), or at any time point (*p* < 0.10) (Figure 2a). The incidence of greater-than-optimal BHBA was also not different between lame and control cows (7.7% vs. 11.1%, *p* = 0.61, Figure 2b). Lame cows had greater overall NEFA concentrations compared to controls (0.32 ± 0.02 vs. 0.26 ± 0.02 mEq/L, *p* = 0.02, Figure 2c) and an overall greater incidence of greater-than-optimal NEFA (53.9% vs. 29.6%, *p* = 0.04, Figure 2d).

### 3.2. Reproductive Performance and Survivability of Lame Cows

During the initial screening (28 to 37 d p.p.), there was a greater proportion of non-cycling lame cows compared to non-lame cows (36.5% vs. 11.1%, resp., *p* = 0.02, Table 1). Of 19 non-cycling lame cows (36.5%), seven (13.5%) were diagnosed with cystic ovarian disease, 11 (21.1%) had repeated follicular waves leading to atresia and one did not have detectable ovarian functions (1.9%). There was no difference regarding non-cycling cows between the two lame groups. The effects of the estrous presynchronization treatment regimen did not differ among the three groups, although cows of group LC responded arithmetically less. Treatment effects on estrous synchronization and ovulation induction were similar between groups. Regarding first-service conception rate (FSCR), non-lame cows tended to differ compared to group LP (*p* = 0.08) and showed numerical difference with the combined lame group (37.0% vs. 21.6%, *p* = 0.14). The values of the other reproductive variables (proportion of Pregnant by 200d, AIs per pregnancy, Days open) did not differ between control and lame groups. Results from the survival analysis indicated there were no differences among cows of the lame and non-lame groups (Figure 3). There, however, were greater for cows of the lame groups (overall 21.2%, *n* = 11) compared to non-lame cows (0%, *p* = 0.01, Table 1). Cullings were mainly related to lameness, infertility, late embryonic loss, or their combination (9 of 11 cases). 

### 3.3. Reproductive Performance and Survivability Based on Energy Status

Regarding metabolic stress expressed through NEFA and BHBA concentrations, only 7 of 79 cows had greater-than-optimal BHBA concentrations whereas 36 cows showed high blood NEFA concentrations during the study period. Based on our results, cows with greater-than-optimal BHBA or NEFA concentrations had no difference (neither statistical nor numerical) regarding reproductive variables compared to cows with optimal metabolic profiles (Table 2). 

Additionally, there was no two-way interaction between lameness and metabolic status regarding any reproductive variable when general linear mixed models were used. 

### 3.4. Reproductive Performance and Survivability Based on Severity and Lesion Type of Lameness

Cows with marked lameness compared to those with moderate lameness had a greater culling percentage (30.6% vs. 0%, *p* = 0.01, Table 3). The values of the reproductive variables were not different among the cows of these groups. In contrast, there was a similar culling percentage for cows with CHDL and ID (21.6% vs. 20%, resp.), but an overall lower reproductive performance. Specifically, CHDL cows had greater than a 25% reduction in FSCR (*p* = 0.04), and 21.8% and 20.3% reductions in the response to estrous presynchronization treatment regimens and in ovulation occurring, respectively (*p* < 0.10). The number of AIs/pregnancy and days open did not differ between the two groups. Results from general linear model analyses indicated there was no two-way interaction for any of the respective variables. Marked lameness was detrimental for survivability irrespective of the type of lesion, and CHDL was detrimental for reproduction irrespective of severity.

## 4. Discussion

To our knowledge, this is the first study to investigate extensively the energy and cyclic status of lame dairy cows and quantify responsiveness to hormonal treatment regimens to induce estrous synchronization. To eliminate the utmost confounding factors, we applied numerous exclusion (health status and parity) and inclusion (days p.p.) criteria, which resulted in a small sample size. Nevertheless, even within this framework, scientific knowledge was obtained.

Regarding the energy status of lame cows, there are inconsistent results reported in the literature. Results from various studies have confirmed that the eating time budgets of lame cows are affected proportionally by the severity of lameness [46,47]. Hence, lame cows can be at a greater risk of NEB. Calderon and Cook [48] reported that there were greater BHBA concentrations in severely to moderately lame cows compared to slightly or non-lame herd mates. Collard et al. [49] reported there was an incrementally greater negative energy balance for cows with compromised locomotion. Conversely, Melendez et al. [8] and Sun et al. [50] reported that there were lower concentrations of NEFA and BHBA, respectively, in lame compared to non-lame cows, which were attributed by the authors to the lower milk yields or to the exhaustion of lame cows. It is probable, that lameness due to its varying severities can lead to variable cow responses. In the present study, due to ethical reasons, we excluded severely lame cows, and as a result we did not expect to find marked effects on cow energy status resulting from lameness. Additionally, cows with a depressed habitus, which would further deteriorate metabolic status, were excluded from the study. As a result, the moderately lower energy status of lame cows in the present study was rather expected and is consistent with the findings of Calderon and Cook [48]. A main finding in the present study was that the suppressed ovarian activity noticed in lame cows was not associated with a lower energy status because there was no interaction when the data for these variables were evaluated. This is consistent with the results from the study of McNally et al. [1], where it was reported that clinically lame cows had a two-fold and a four-fold greater risk of NEB and anestrus respectively, yet no relationship between the latter conditions was evident. 

The suppressed ovarian cyclicity of lame compared to non-lame cows by the end of puerperium (36.5% vs. 11.1%) was mainly attributed to a greater incidence of cystic and atretic ovarian follicles. These findings are consistent with results from various studies where it was reported that there was an increased risk of anovulation [51], delayed onset of estrous cycles [52,53] and a greater incidence of cystic ovarian disease [9] in lame cows compared to non-lame herd mates. Lameness as a major source of inflammation [54,55,56] can directly affect reproductive performance. Interleukin-1 alpha is found to inhibit pre-ovulatory surge releases of LH and interleukin-2 has direct functions in peptide release from the anterior pituitary gland [57]. Moreover, lameness scoring has been correlated with hyperalgesia and haptoglobin values [54], whereas increased haptoglobin concentrations have been related to post partum anovulation in dairy cows [58].

Even though lameness has been associated with anovulation, research on the use of hormonal protocols to alleviate this effect is scarce. In the present study, both hormonal treatment regimens for the presynchronization and synchronization of ovulation were adequate for initiating ovarian functions (i.e., formation of a functional CL afterwards), since almost 90% of lame cows were synchronized and 80% ovulated. Arithmetically, the P4 protocol seemed to benefit cows in terms of presynchronization success and the proportion of non-cycling cows that ovulated; however, the FSCR was lower compared to that under the double-PG protocol. Further studies are necessary to confirm these findings. Conversely, lame cows had a lower FSCR compared to control cows (21.6% vs. 37%), although this finding did not reach the significance level (*p* = 0.14). In the study of Melendez [9], non-lame cows were 4.22 times more likely to conceive, with the FSCR being similar to those in the results in the present study (42.6% vs. 17.5%, *p* ≤ 0.05). These findings are indicative that there could have been compromised oocyte quality and/or an increased likelihood of early embryonic death in lame cows. Progesterone deficiency is a common causal factor for both these conditions [59,60,61] and has been linked to subfertility when occurring in the periovulatory period [62,63,64]. In various studies, lame cows had lower progesterone values compared to non-lame cows. There have been reports of suboptimal progesterone concentrations in lame cows in previous studies: (i) during the 6 days before estrus [10], (ii) 12 to 17 d post-ovulation [14], and (iii) continuously between 30 to 64 d p.p. [8]. Progesterone supplementation has been used in estrous synchronization regimens to improve conception and embryonic survival [65,66,67] and to treat ovarian cysts [68] or non-functional ovaries [69]. McNally et al. [1] used progesterone treatment to synchronize estrus in cows with underlying diseases, including lameness. Similarly to findings in the present study, no difference between lame and non-lame cows was evident regarding responsiveness to the hormonal treatment regimens, yet only 22.7% of lame cows vs. 53.9% of non-lame ones conceived [1]. 

Although it was not a primary aim in the present study, long term reproductive performance was similar between lame and non-lame cows. Through careful evaluation of the Kaplan–Meier plot (Figure 3), there was initially a greater percentage of cows becoming pregnant in the non-lame group (50 to 100 DIM); however, there was decrease in pregnancy percentage in the lame group from 100 to 150 DIM, when the lameness condition had likely subsided in severity in most of the cows. As a result, long-term reproductive indices were similar in lame and non-lame cows, which probably makes lame cows good candidates for applying an extended voluntary waiting period. Only a few studies have been conducted to evaluate reproductive indices through 100 to 150 DIM (apart from the FSCR) or have included estrous synchronization treatment regimens in lame cows. Based on previous findings, there were 12 to 70 more days to conception and 0.2 to 2 more AIs for lame cows to become pregnant compared to non-lame herd mates [70,71,72,73,74]. Lameness also suppresses behavioral signs of estrus, with more than 30% of lame cows having relatively short durations of behavioral estrus compared to 18% non-lame cows having these [15]. Considering these outcomes, reproductive management procedures including treatment of lame cows to induce the onset of estrous cycles or treatment schemes for anovulation early in the lactation period can be beneficial, especially if lameness is treated promptly. 

A major consideration regarding lame cows was the greater-than-acceptable culling rates in this group, which were attributed primarily to lameness and/or infertility. Dairy management for culling is usually multifactorial; nonetheless, in a recent study, lameness was the main reason for 8.4% of cullings [75]. Notably, in the present study, the severity of lameness was the most informative factor regarding culling rates. In this context, moderately lame cows differed from cows with marked lameness. Similarly, Bicalcho et al. [71] used visual locomotion scoring (VLS) to assess the severity of lameness and reported that a visual locomotion score of ≥ 3 or ≥ 4 increases the hazard ratio of culling by 1.45 and 1.74, respectively, compared to cows with a lower visual locomotion score. These findings are indicative of the need for the prompt diagnosis and treatment of lameness cases before a deterioration in the lameness score (and in most cases in the integrity of deeper structures of the claw) occurs. From the studies of Hernandez et al. [73], Leach et al. [76] and Gundelach et al. [77], it is apparent that the negative impact of lameness on various aspects of dairy production can be alleviated if there is a lower threshold for the detection of these conditions.

The type of lesion was much more informative regarding reproductive efficiency, because cows with CHDL had an overall lower performance compared to cows with ID. Actually, the reproductive indices of cows with ID in the present study were equal to those of non-lame cows (i.e., the FSCR was 37% vs. 40%, and the percentage of cows pregnant by 200 d was 88.9% vs. 92.9% for non-lame vs. lame ID cows, respectively), although at the initiation of the study the proportion of anestrus was greater in the ID cows. Additionally, severity, expressed through the lameness score and number of claws afflicted, did not differ between the CHDL and ID groups (67% and 65% severe cases, respectively). Hence, the lower fertility of lame cows in the present study can be solely attributed to cows with CHDL. In fact, based on the findings in the present study (data not shown) and on other bibliographical reports, clinical healing is much more delayed (by up to 8 weeks) in cases of CHDL compared to ID, the latter comprised mostly of digital dermatitis lesions [78]. Also, consistently with findings in the present study, overall reproductive performance was greater in cows with lesions attributed to infectious causes than in those with disruptions of the horn [70,79]. In the pioneering study of Morris et al. [14], almost half of the lame cows responded well to a GnRH ovarian stimulation program, while the other half of the cows were non-responsive to this treatment, with responses differing from non-lame cows. These findings emphasize the need for more focused studies on the effect of the type, severity, and probably the chronicity of lesions on reproduction. 

## 5. Conclusions

Based on the results of the present study, we conclude that cows that are lame at the end of puerperium experience both suppressed ovarian activity and energy status; however, these conditions are not necessarily related. The implementation of estrous presynchronization and synchronization treatment regimens in lame cows was effective at inducing ovarian functions and partially at the restoration of long- but not short-term reproductive functions. The severity of lameness was a primary factor influencing the probability of culling, whereas compromised reproduction was mainly attributed to cows with claw horn disruption lesions. 

## Figures and Tables

**Figure 1 animals-13-01537-f001:**
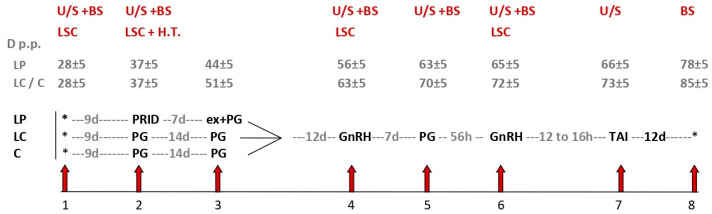
Experimental design-schematic representation of interventions. D p.p. = days post partum; LP/LC/C = groups Lame-Progesterone, Lame-Control and Control, resp.; U/S = transrectal ultrasonography; BS = blood sampling; LSC = lameness scoring; H.T. = hoof trimming; * = Time points (onset and termination of experimental protocol); PRID = progesterone-releasing device; ex = removal of PRID; PG = administration of prostaglandin F_2α_; GnRH = administration of gonadotropin releasing hormone; TAI = timed artificial insemination.

**Figure 2 animals-13-01537-f002:**
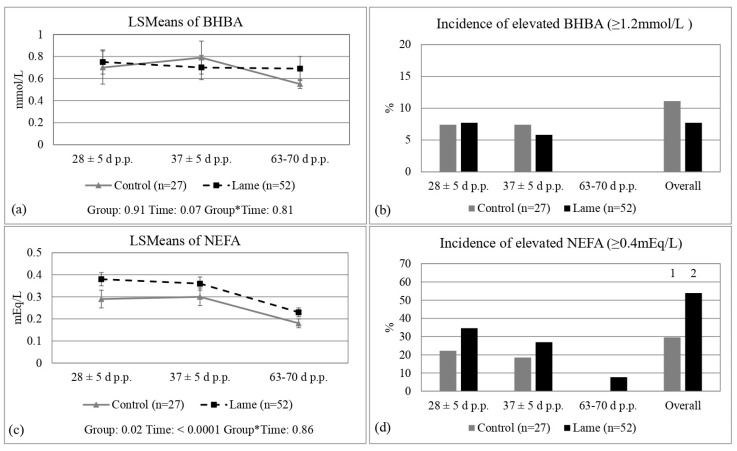
Least-square means (±SE) and incidence of elevated ß-hydroxybutyric acid (BHBA, (**a**,**b**), resp.) and non-esterified fatty acid (NEFA, (**c**,**d**), resp.) concentrations in control and lame cows at 28, 37, and 63 to 70 days post partum. ^1,2^: Different numbers denote differences (*p* < 0.05), *: Denotes interaction term.

**Figure 3 animals-13-01537-f003:**
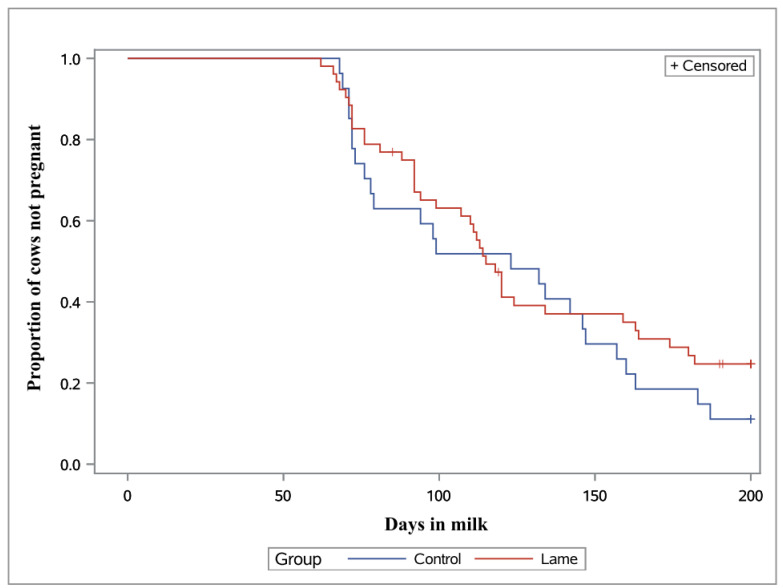
Kaplan–Meier survival curves for the proportion of non-pregnant cows in the Control and Lame groups. Cows that were culled and that conceived after 200 DIM are censored (*p* = 0.31).

**Table 1 animals-13-01537-t001:** Reproductive variables and culling rate of control (group C) and lame cows when there was use of two presynchronization treatment regimens (groups LC and LP).

	Group			
Variable	C	LC	LP	LC + LP
n	27	26	26	52
Non-cycling (%)	11.1 ^a,1^	38.5 ^b^	34.6 ^b^	36.5 ^2^
Presynchronization success (%)	81.5	65.4	76.9	71.2
Synchronization success (%)	92.6	84.6	92.3	88.5
Ovulation (%)	81.5	80.8	76.9	78.9
Non-cycling ovulation (%)	100	70	77.7	73.7
FSCR (%)	37.0	26.9	16.0	21.6
Pregnant by 200 d (%)	88.9	70.8	87.5	79.2
AIs/pregnancy (mean ± SE)	3.1 ± 0.4	2.8 ± 0.5	3.0 ± 0.5	2.9 ± 0.3
Days open (median)	123	114	111	112
Culled (%)	0.0 ^a,1^	23.1 ^b^	19.2 ^b^	21.2 ^2^

^a,b:^ Different letters denote differences between Control, Lame—Control and Lame—Progesterone groups (*p* < 0.05). ^1,2^: Different numbers denote differences between Control and Lame (LC + LP) groups (*p* < 0.05).

**Table 2 animals-13-01537-t002:** Reproductive variables and culling rate of cows with greater-than-optimal ß-hydroxybutyric acid (BHBA+) and non-esterified fatty acid (NEFA+) concentrations during the study period.

	Group			
Variable	BHBA−	BHBA+	NEFA−	NEFA+
n	72	7	43	36
Non-cycling (%)	27.8	28.6	27.9	27.8
Presynchronization success (%)	73.6	85.7	72.1	77.8
Synchronization success (%)	90.3	85.7	88.4	91.7
Ovulation (%)	79.2	85.7	79.1	80.6
Non-cycling ovulation (%)	80.0	50.0	83.3	70.0
FSCR (%)	26.8	28.6	27.9	25.7
Pregnant by 200 d (%)	83.8	71.4	82.9	82.4
AIs/pregnancy (mean ± SE)	2.9 ± 0.3	3.6 ± 1.1	3.2 ± 0.4	2.7 ± 0.4
Days open (median)	112	134	115	109
Culled (%)	15.3	0.0	14.0	13.9

**Table 3 animals-13-01537-t003:** Reproductive variables and culling rate of cows with moderate or marked lameness and with lesions due to claw horn disruptions (CHDL) or infectious diseases (ID).

	Group		Group	
Variable	MODERATE	MARKED	CHDL	ID
N	16	36	37	15
Non-cycling (%)	25.0	41.7	40.5	26.7
Presynchronization success (%)	75.0	69.4	64.9	86.7
Synchronization success (%)	93.8	86.1	83.8	100
Ovulation (%)	81.3	77.8	73.0	93.3
Non-cycling ovulation (%)	75.0	73.3	66.7	100.0
FSCR (%)	25.0	20.0	13.9 ^1^	40.0 ^2^
Pregnant by 200 d (%)	81.3	78.1	73.5	92.9
AIs/pregnancy (mean ± SE)	3.0 ± 0.5	2.8 ± 0.5	3.0 ± 0.4	2.6 ± 0.6
Days open (median)	115	111	114	110
Culled (%)	0.0 ^a^	30.6 ^b^	21.6	20.0

^a,b^: Different letters denote differences between the groups with moderate and severe lameness (*p* < 0.05). ^1,2^: Different numbers denote differences between the groups with claw horn disruption lesions (CHDL) and infectious diseases (ID) (*p* < 0.05).

## Data Availability

Data are available upon request to the authors.

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
