# Peer review of "Ovarian and Energy Status in Lame Dairy Cows at Puerperium and Their Responsiveness in Protocols for the Synchronization of Ovulation"

_animals, 2023, doi:10.3390/ani13091537_

Round 1

Reviewer 1 Report

Ovarian and energy status in lame dairy cows at puerperium and their responsiveness in protocols for the synchronization of ovulation

This paper presents the results of an experiment conducted on lame cows at the end of puerperium. Specifically, the authors set out to investigate on the one hand the relationship between lameness and energy status, and on the other hand, the physiological response at the ovarian level in general and with respect to different hormonal synchronization protocols. The main contributions are the observation that the interactions between lameness, energy status, and resumption of ovarian activity after parturition are less linear than previously believed. In addition, a prompt diagnosis of the main lesion causing lameness is required, since severity of the lesion influences successive performance and culling rate. This confirms that proper herd health management must necessarily include a careful clinical examination of single animals, especially in a delicate phase such as puerperium.

This work is scientifically sound and very informative. Despite the impressive amount of work that the experimental part conceivably required, both the descriptive parts and result/discussion are presented in a scientifically detailed and transparent, yet easy to follow manner.

Minor comments:

Line 140non-lame cows: Did all non-lame cows in this study have a locomotion score of 1 by Sprecher et al., or were cows with score 2 also included? In the reviewer’s opinion, both situations would be acceptable for the purpose of the experiment, but this detail is relevant information. (Incidentally, it could be an idea for a future study, to repeat the procedures clustering cows with very mild lameness into a separate group).  

Line 149: after disinfection of the vulva: I suggest indicating more specifically how the vulva was disinfected.

Line 210 and following (versus for example line 263): Please check the manuscript for uniform use of spaces before and after mathematical characters such as “<” and “>”.

Line 350: lower milk yields: It is clear that severely lame cows had to be excluded from your study and, arguably, those animals yielded less milk as well. It would be interesting to know about the milk yields of the animals in your experiment specifically. Is there any information available? If so, I would advise considering to integrate it in the results and discuss it briefly in this paragraph. Regarding the relationship between the resumption of ovarian activity and the energy status of high-yielding dairy cows, and in case the authors see fit to emphasize this topic in the discussion, they might also consider mentioning the much-discussed prolongation of the voluntary waiting period (extended lactation).

Author Response

Please see in the attachment.

Reviewer 2 Report

Line 57- Is there a difference between Acute vs Chronic Lameness on infertility? What is the difference between these two classifications? 

Line 63- Is this in acute or chronic lameness? 

Line 79- Is severe lameness acute, chronic or both? 

Line 99- Was lameness defined between acute and chronic? What was the distinction between these two classifications? 

 Line 123- The different interventions are not clear and provides confusion to the reader. 

Line 135- Were both acute and chronic lameness used for this study?

Line 147- Why was there not a control progesterone group? The authors need to provide some explanation as to why this was not performed. 

Line 200- What type of images were taken from the ultrasound and stored? This is unclear and needs more information. 

Figure 2- Provide letters to demonstrate statistical significance on graphs, as it is hard to determine on the graph between the text. 

Table 1 and 2- P-values would be helpful to be listed within the table. 

Line 334- A two-way interaction is not an appropriate statistical model since the same animal would be in two different groups. 

Table 3- Break this into 2 separate tables with p-values, as having them together makes it appear that the four groups were compared against each other. 

No major issues with the quality of English. 

Author Response

Please see in the attachment.

Reviewer 3 Report

General Comments

The research reported in this manuscript is not especially novel. The study was conducted appropriately to address the objectives. The greatest problem is the written quality of the manuscript. This reviewer has dedicated a large amount of time to providing edits to improve the written quality of the manuscript and these are included in the attached PDF file. There may be a few problems with this reviewers edits because of lack of time to carefully review the edits before the due date for this peer review, however, the authors should be able discern any errors in the reviewers feedback and make appropriate changes.

Specific Comments

Detailed edits are included on the attached manuscript along with some specific comments regarding clarity of communication.

Reviewer 4 Report

The manuscript deals with a study investigating the association between the ovarian and energy status of multiparous lame cows at the end ot the puerperium and futhermore, to investigate their responsiveness to synchronisation protocols. A total of 79 Holstein Frisian cows were included in the study. The influence of lameness on reproductive performance and its therapeutic influence are of relevance to dairy sector.

In general, the work is well written and easy to read and understand. The title reflects the scope of the work. The introduction is of reasonable length and provides a rationale for the study. The objective is clearly stated, and the work was hypothesis-driven. The statistical models used to analyze the data are appropriate. The results are mostly clearly presented. The discussion is well organized, and the references used to discuss the results are appropriate.

However, I have a major concern about the small number of study animals. This aspect was also mentioned self-critically by the authors (see L341). In general, much higher numbers of cases are needed to show differences in reproductive performance. In this context, I ask the authors to provide a valid sample size analysis in the Materials and Methods section. What assumptions werde made?
Furthermore, I consider the presentation of the results as percentages critical, since e.g. one animal of the control group (n=26) makes a difference of approx. 4 percentage points.

I have further concerns because of the long study period of approximately 3.5 years and the 3 study farms used. Can you explain any other background for the long duration of the study, which has already ended in 2019? How could it be excluded that with the small number of animals, farm management procedures, feeding, climatic conditions, etc. have an influence on the study results?

Specific comments:

L113-120 Where the milk production, housing and feeding conditions comparable in all 3 farms? Were the study animals per treatment group evenly distributed on all 3 farms?

L114-115 The study was conducted over a period of 3.5 years. How was a possible influence of time taken into account, e.g. regarding climatic influences, feed quality, etc?

L128 Does the weekly examination mean they were free of any lameness until day 28±5?

L136 How much was the calving date allowed to differ in blocked animals?

L341 In order to be able to judge whether ‘useful conclusions’ can be drawn, a valid sample size analysis is necessary.

Round 2

Reviewer 3 Report

General Comments

The authors effectively responded to this reviewers feedback with the most recent submission.

Reviewer 4 Report

The authors have taken into account my correction requests and comments. The quality of the manuscript has increased significantly. Thank you!